# The Neural Development of Chinese Lexical Tone Perception: A Mismatch Negativity Study Across Childhood, Adolescence, and Adulthood

**DOI:** 10.3390/brainsci15010093

**Published:** 2025-01-19

**Authors:** Han Wu, Yixiao Zhang, Yiru Liu, Shijun Zhang, Linjun Zhang, Hua Shu, Yang Zhang

**Affiliations:** 1Institute on Education Policy and Evaluation of International Students, Beijing Language and Culture University, Beijing 100083, China; wuh2017@blcu.edu.cn (H.W.); 202321198055@stu.blcu.edu.cn (S.Z.); 2School of Biomedical Engineering, Dalian University of Technology, Dalian 116038, China; yixiao.zhang@donders.ru.nl; 3International Division, The Second High School Attached to Beijing Normal University, Beijing 100088, China; 11112019053@bnu.edu.cn; 4School of Chinese as a Second Language, Peking University, No. 5 Yiheyuan Road, Haidian District, Beijing 100871, China; 5State Key Laboratory of Cognitive Neuroscience and Learning, Beijing Normal University, Beijing 100875, China; shuh@bnu.edu.cn; 6Department of Speech-Language-Hearing Sciences and Center for Neurobehavioral Development, University of Minnesota, Minneapolis, MN 55455, USA; zhang470@umn.edu

**Keywords:** mismatch negativity (MMN), lexical tone perception, development, childhood, adolescence

## Abstract

Background/Objectives: In a tonal language like Chinese, phonologically contrasting tones signify word meanings at the syllable level. Although the development of lexical tone perception ability has been examined in many behavioral studies, its developmental trajectory from childhood to adulthood at the neural level remains unclear. This cross-sectional study aimed to examine the issue by measuring the mismatch negativity (MMN) response to a Chinese lexical tonal contrast in three groups. Methods: The MMN response to a flat-falling tonal contrast (Tone1 versus Tone4) were recorded from children (25 participants aged 6–8), adolescents (26 participants aged 12–14), and young adults (20 participants aged 18–24). Nonsense speech stimuli were also used by superimposing Tone1 and Tone4 on an English syllable. Results: All three groups demonstrated typical early MMN responses in both the meaningful and nonsense syllable conditions. However, the MMN amplitudes varied significantly across groups, with the child group showing smaller responses compared to the adolescent and adult groups, while the latter two groups had similar MMN amplitudes. Conclusions: Neural sensitivity to tonal contrasts is not fully mature in children and reaches a more adult-like level during adolescence, with no significant difference in sensitivity to meaningful versus nonsense syllables. These results provide new insights into the neural development of lexical tone perception in a tonal language, highlighting its maturation during adolescence in this process.

## 1. Introduction

The study of speech perception provides an important entry point for understanding how humans acquire their native language. Research has shown that infants gradually improve their ability to recognize native language sounds and features, while their ability to discriminate nonnative language features declines [1,2,3,4,5]. This preference for the native language sounds in infancy can, to some extent, predict the development of future reading skills [6,7]. However, speech perception does not fully mature during infancy. Instead, it continues to develop over time, influenced by their linguistic environment and the maturation of the auditory system [6,8,9,10,11]. Additionally, extensive reading and writing experiences can further enhance speech perception ability [12,13]. For example, even at age 12, children have not yet reached the level of categorical perception seen in adults and are not able to use fine-grained spectral cues as adults do [14]. The precise age at which speech perception fully matures remains unclear [15,16]. Moreover, most previous studies have focused on non-tonal languages, leaving limited evidence on the development of lexical tone perception.

Behavioral and electrophysiological measures are most commonly used in speech perception research. Despite the task’s simplicity, it is often challenging to use behavioral methods to examine speech perception in children. Compared to adults, children are less likely to maintain focus on a task for an extended period of time and their motivation or patience may easily influence the task outcomes. Therefore, it is unclear whether behavioral improvements should be interpreted as an increase in the capacity to perceive speech sounds or as the ability to use strategies to complete a task. By contrast, the electrophysiological method is an objective technique for recording the brain’s activity in response to the stimuli. Specifically, the mismatch negativity (MMN) response, a well-known component of event-related potentials (ERPs), is widely used to investigate speech perception in infants, children, and adults [17,18,19,20,21,22,23]. MMN is a difference waveform obtained by subtracting the standard stimulus from the deviation stimulus in an oddball paradigm. As the MMN component reflects automatic detection of acoustic and phonological changes during speech perception, it is particularly suitable for testing speech perception in infants and children who are not required to engage in an active listening task [24].

Some previous studies have utilized MMN to examine the developmental aspects of speech perception [25,26,27,28,29,30,31]. For example, Shafer et al. measured the development of mismatch responses (MMRs) to the English vowel contrast (/I/ in bit versus /e/ in bet) in children from 4 to 5 and 6 to 7 years old [32]. Their findings demonstrated that in the early time window (200–300 ms), most of the younger children showed positive MMRs (i.e., p-MMRs), whereas more than half of the older children showed negative responses. Moreover, there was no significant difference in the later time window (300–400 ms) between the two groups of children. Bishop et al. examined MMNs elicited by the change from /ba/ to /da/ in children (7–12 years), teenagers (13–16 years), and adults (35 to 56 years) [33]. Their findings showed that in the early time window (100–250 ms), the amplitude of MMN increased with age, while in the late time window (300–550 ms), the amplitude of the late discriminative negativity (LDN) decreased with age.

Although there are numerous studies on the development of consonant and vowel perception, only a small number of studies have examined the development of Chinese lexical tone perception [16,34]. For example, Ref. [16] found that a large tonal difference (Tone1 versus Tone3) elicited typical MMNs in the 150–300 ms time window in all the three groups of children (i.e., 4, 5, and 6 years old), whereas a small tonal difference (Tone2 versus Tone3) did not elicit the typical MMN response but only P-MMRs in the 5- and 6-year-old groups. Liu et al. also showed that a small tonal difference (Tone2 versus Tone3) did not elicit the typical MMN responses but only elicited LDN responses in preschool (3–4 years) and school-aged children (7–8 years) [34]. No conclusive results are available on the development of lexical tone perception from childhood to adolescence and adulthood as only children were included in the previous studies.

To distinguish acoustic processing from phonological processing during speech perception, comparisons could be made between native versus nonnative speech sounds and between speech versus nonspeech sounds [23,35,36,37,38]. For example, Näätänen et al. compared distinct phonological versus acoustic processing by using a Finish vowel contrast and a Finish–Estonian vowel contrast [36]. Vandermosten et al. created nonspeech sounds by rotating the frequencies of the second formant transition of consonants for comparing speech and nonspeech processing [37]. Various acoustic manipulations have also been applied to lexical tones, but the results showed clear evidence of transfer of learning regardless of whether the tones were carried by native, nonnative, and nonspeech sounds [23,38,39,40]. For example, Xu et al. demonstrated that when native Thai speakers discriminated Thai tones that were superimposed on a Chinese syllable (so-called tonal chimeras), the left planum temporale was activated, indicating that phonological processes were recruited [38]. It remains to be tested whether tonal contrasts in the context of tonal chimeras and naturally produced lexical tones would be similarly perceived in children. In a recent article by Stilp et al., the researchers discussed the importance of considering the faithfulness of nonspeech stimuli to speech acoustics, their role within everyday hearing, and their relationship to listening experience and recognition [41]. In this regard, using chimeric lexical tones in a nonnative speech context provides a middle-ground approach to the study of phonological vs. acoustic processing. This approach adds a different acoustic control method from traditional comparisons using native vs. non-native or speech vs. nonspeech as it allows researchers to isolate the perceptual processes involved in recognizing identical tonal patterns with and without the influence of native phonological context. It can be adopted to provide insights into the developmental trajectory and maturation of tonal perception, which can reveal how children and adults may use similar or different processing strategies for native and nonnative speech signals carrying the same lexical tone contrast.

The current study aimed to examine the development of lexical tone perception at the neural level, which is reflected by the electrophysiological index (i.e., the MMN response) from childhood to adulthood. We recorded MMN responses to a Chinese tonal contrast (Tone1 versus Tone4) in children (6–8 years), adolescents (12–14 years), and young adults (aged 18–24 years), with the two lexical tones in a natural Chinese syllabic (natural tones, NT) context and a chimeric English syllabic (chimeric tones, CT) context, respectively. We predicted that typical early MMN responses would be observed in all the three groups and that the NT and CT contrasts would elicit similar MMN responses in the adult group irrespective of the syllable context. However, we were unable to make specific predictions about the developmental trajectory of MMN responses from childhood to adolescence and adulthood, or whether the contrasts in the NT and CT contexts would elicit similar MMN responses in the child and adolescent groups.

## 2. Materials and Methods

### 2.1. Participants

Three groups were included: Group 1, 25 children aged 6–8 years (mean age = 92.4 months, SD = 8.2); Group 2, 26 adolescents aged 12–14 years (mean age = 156.0 months, SD = 8.1); and Group 3, 20 young adults aged 18–24 years (mean age = 254.4 months, SD = 17.8). We used G*Power (Version 3.1.9.7) to verify a priori sample size requirement for MANOVA with three subject groups and six within-subject measures (three electrodes and two syllable contexts) [42]. To achieve a relatively large effect size of 0.4 at the alpha level of 0.05 and the power of 0.8 with our experimental setup, the required total sample size would be 56. Our total sample size of 71 exceeded the required minimum.

All participants had normal hearing with no history of neurological diseases or speech/language disorders. This study was approved by the ethics review board at Beijing Language and Culture University (approval code: #2019-11-01). Informed consent was obtained from all of the adult participants and parents of the child participants.

### 2.2. Stimuli

Two sets of stimuli were constructed for this experiment by superimposing two Chinese lexical tones (i.e., Tone1 and Tone4) on the Chinese syllable /tɕi/ and English syllable /vi/. The original stimuli were recorded at a sampling rate of 44.1 kHz from a male Chinese–English bilingual speaker. The two monosyllables were digitally edited with duration (200 ms) and intensity (70 dB SPL) equalized. Pitch tier transfer was then performed using the Praat software (http://www.fon.hum.uva.nl/praat/, accessed on 1 December 2024) to superimpose pitch patterns of the two lexical tones on the two syllables while keeping the rest of the acoustic features identical. This procedure generated two pairs of stimuli, i.e., /tɕi1/ versus /tɕi4/ and /vi1/ versus /vi4/ (Figure 1).

### 2.3. ERP Procedure

Four oddball blocks were presented to each participant with each syllable condition consisting of two blocks in which the standard and deviant stimuli were counterbalanced. Each block began with 30 standard trials, followed by 723 trials comprising 87.5% standards and 12.5% deviants. The order of stimuli was pseudo-randomized with at least three successive standards between deviants. The interstimulus interval was 450 ms. The participants were seated on a comfortable chair in a shielded chamber. They were instructed to watch a selected animation movie but to ignore the sounds. The movie was presented in a DVD player, which was set approximately 60 cm in front of the seat. Participants were informed to complete a questionnaire about the movie after the experiment.

### 2.4. EEG Recording and Data Analysis

The 128-channel EEG signals were recorded using a Hydrocel Geodesic Sensor Net (Electrical Geodesics Inc., Eugene, OR, USA) referenced to Cz. Data were sampled at 500 Hz/channel with filters set at 0.01–100 Hz and calibrated technical zero baselines. Electrode impedances were maintained below 50 kΩ during the recording.

Off-line analysis of EEG signals was re-referenced to the average of all scalp channels. Signals were digitally filtered with a 0.3–30 Hz band-pass filter using Netstation software (Version 4.2). The filtered EEG data were first submitted to BESA for bad channel interpolation. Specifically, data from channels with excessive artifacts were spline-interpolated (BESA 5.1 software) and artifacts exceeding ±100 µV in any channel were automatically rejected. Independent component analysis (ICA) was then performed in EEGLAB to remove eye blinks and eye movements. Epochs were set at 600 ms in length, including a 100 ms pre-stimulus baseline.

Average ERPs were calculated separately for standard and deviant stimuli in the two syllable contexts for each participant, and difference waveforms were produced by subtracting the standard waveform from the deviant waveform for every participant. As the literature review in the Introduction shows, MMN quantification using subtracted ERP waveforms would face the polarity mixture problem, particularly for the child and adolescent groups. This issue arises because the presence of both MMN and P-MMR responses within these age groups can create variability in the polarity of the waveforms, making it difficult to accurately quantify and compare the data across individuals. There are several strategies to address this problem for statistical comparison between groups. One approach is to separate subgroups for analysis to ensure that each subgroup’s data can be evaluated independently. Alternatively, researchers have attempted to rectify the polarity discrepancies among subjects using methods such as global field power measures [43,44] or time-frequency analysis [33,44,45], which involves analyzing spectral power or inter-trial phase coherence using phase-locking values. Another technique involves temporal principal component analysis (t-PCA) [46,47], which can realign the polarity of selected principal components for the target MMN/p-MMR activity to facilitate between-group comparison. By applying PCA to the time series data, researchers would have the opportunity to evaluate the components that maximally account for the variance in the ERP data and isolate the MMN response from overlapping brain activities to enhance the reliability of MMN amplitude and latency quantification based on its temporal and spatial patterns [48].

In our t-PCA analysis, a fourth-order tensor (electrodes × time points × stimuli × subjects) was constructed and imported into the analysis toolbox [49]. Here, “electrodes” refer to the EEG recording channels, “time points” correspond to the sampled intervals of the averaged ERPs, “stimuli” represent the two syllable contexts, and “subjects” encompass the children from the three experimental groups. The dataset was first filtered using the wavelet filter, and the selected mother wavelet is the reverse biorthogonal wavelet of the order of 6.8 (rbio6.8), which is ideal for decomposing conventionally averaged EEG recordings of children’s MMN activity elicited by an uninterrupted sound using the oddball paradigm [50]. In order to extract the component of interest, t-PCA and Promax rotation were applied. Then, the data were projected to the electrode field to correct the indeterminacy of variance and polarity [49]. Finally, the amplitude of the MMN response for three electrodes sites of interest (Fc3, Fc4, and Fcz) on the time window was extracted and the topographies were plotted. Based on the final grand average waveforms in this study, and the time window of MMN in the previous studies [51,52], we chose 30 ms wide windows placed 230–260 ms for the NT context and 250–280 ms for the CT context. A repeated measures MANOVA was performed to examine the difference in MMN responses, with group (group 1, 2, 3) as the between-subject factor and electrode (Fc3, Fc4, Fcz) and syllable context (NT and CT) as the within-subject factors.

## 3. Results

During the procedure of t-PCA, 16 components that explained 99% of the variance were retained. Based on the temporal and spatial properties of the MMN response and the similarity of spatial components among all the participants, the first and eleventh components, which, respectively, peaked at 274 ms and 236 ms with topographical distributions that characterized the MMN response, were selected. These components explained 51.42% and 1.14% of the variance, respectively (Figure 2). Although the polarity of the projected first and eleventh components was positive when viewed in the PCA space, they could be considered negative components because their original waveforms were negative-going peaks [53]. This happens because PCA maximizes variance and can rotate components in a way that changes their apparent polarity. By definition, variance is always non-negative. In PCA, the direction of the components can be reversed without affecting the variance they explain. As a result, components with negative peaks in the original signal can appear positive after PCA simply because of the reorientation of the components during the transformation process. However, when these components are back-projected into the electrode space, their spatial patterns align with their original negative peaks, revealing their true negative polarity. Thus, the polarity of the target components can be consistently realigned across subjects to resolve issues related to the polarity mixture problem in the mismatch responses.

The difference waveforms across the three age groups for the two syllable conditions and three electrodes, and the corresponding brain topographies, are shown in Figure 3. The results of the repeated measures MANOVA demonstrated that the main effect of the group was significant (*F* (2, 68) = 8.631, *p* < 0.001), but the main effects of electrode (*F* (2, 68) = 1.520, *p* = 0.222) and syllable context (*F* (1, 68) = 0.803, *p* = 0.373) fell short of significance. Neither the two-way interactions nor the three-way interaction was significant (all *p*s > 0.05). Pairwise comparisons were applied as post hoc tests for the significant main effect of group. The results showed that the difference between the children and adolescent groups (Bonferroni corrected *p* = 0.030) and the difference between the children and adult groups (Bonferroni corrected *p* < 0.001) were significant.

## 4. Discussion

The present study investigated the development of MMN response to the Chinese lexical tonal contrast from childhood to adolescence and young adulthood and aimed to clarify whether the development pattern was affected by the native (i.e., NT) versus nonnative (i.e., CT) syllable context on which the lexical tones were superimposed. Our results revealed a significant difference in the MMN amplitude between the child and adolescent groups but a nonsignificant difference between the adolescent and adult groups, indicating that lexical tone perception develops continuously in childhood and does not fully mature until adolescence. Moreover, the same developmental pattern was observed in both NT and CT conditions, indicating that phonological knowledge of lexical tones is easily transferrable and affects the perception of pitch contours in the nonnative and nonsensical syllabic context.

Speech perception continues to develop from birth influenced by both the language environment and biological maturational changes to the auditory and cognitive systems [54]. However, the form and complexity of behavioral tasks make it challenging to track the development of children’s speech perception skills. For example, young children are less adept than older children and adults at focusing on long and complex sequences of sounds and often lack the motivation and sustained attention to follow the task instructions. Consequently, it is often unclear in behavioral studies of speech perception whether poorer performance in children is due to inadequate speech perception or the underdevelopment of cognitive systems related to executive processing (e.g., decision making) [55]. Electrophysiological measures of brain responses are valuable for clarifying the source of children’s speech perception performance, as they allow the examination at levels preceding behavioral responses. In particular, a great number of studies have adopted the oddball paradigm and ERP components (e.g., MMN and P3a) to examine speech perception by children of varying ages [56,57,58,59,60,61,62,63,64,65].

Previous findings have demonstrated that the MMN response is sensitive to a variety of acoustic and phonological contrasts. However, it remains unclear whether the typical early MMN is present in young children and how maturation influences the amplitude and topography of MMN [16,32,34,61,66,67,68,69]. In the present study, the typical early MMN response was observed in all the three groups. This result contrasts with previous research indicating that only the adults showed the typical early MMN in response to a Chinese tonal contrast, while two groups of children (preschool- and school-aged) showed negative responses only in the later time window [34]. This discrepancy might be partly due to differences in the tonal contrasts used in the two studies: Tone1 versus Tone4 has a large tonal difference in the present study, and Tone2 versus Tone3 involves a relatively small tonal difference in Liu et al.’s study. Interestingly, a recent study found that in preschool-aged children, the lexical tonal contrast with a large tonal difference (Tone1 versus Tone3) elicited the typical early MMN whereas the tonal pair with a small tonal difference (Tone2 versus Tone3) did not produce significant MMNs in either the early or late time windows [70]. Furthermore, the results of the present study showed that the amplitude of MMN in the adolescent group was significantly larger than in the child group, revealing for the first time that the ability to discriminate Chinese lexical tones (at least those with a large tonal difference) is an ongoing process from childhood to adolescence.

MMN is sensitive to both acoustic and phonological changes with phonological changes eliciting larger MMNs than equivalent acoustic changes [23,36]. In the present study, the detection of change in pitch contours elicited similar MMNs irrespective of whether the pitch contours were superimposed on Chinese syllables to form Chinese lexical tones or on English syllables to form tonal chimeras. This finding indicates that the phonological knowledge of lexical tones is easily transferrable and affects pitch perception. Our results are consistent with previous findings that native Chinese speakers’ perception of pitch in nonnative speech and even nonspeech sounds is influenced by their linguistic experience with lexical tones [23,38,39,40]. For example, Bent et al. found that native language experience affected the identification of nonspeech pitch contours [39]. Specifically, native Chinese speakers were more likely to misidentify flat and falling pitch contours than native English speakers and these misidentifications were linked to the specific features of Chinese lexical tones. In a study by Xu et al., activation of the left planum temporale, a brain region associated with phonological processing, was observed when native Thai speakers discriminated Thai tones superimposed on a Chinese syllable (tonal chimeras) [38]. The present study further indicates that lexical tone knowledge affects the perception of pitch contours in nonnative speech sounds, even in children below the age of 8 years old. These results underscore the adaptability and transferability of linguistic knowledge. Children who are exposed to and learn tonal language patterns can apply this knowledge to recognize pitch variations in nonnative speech, such as in tonal chimeras. The ability to transfer phonological knowledge to nonnative contexts suggests that early linguistic experiences shape not only the perception of native speech but also how children process and interpret unfamiliar or modified speech signals. This ability to generalize tonal processing is significant as it highlights the far-reaching impact of early language exposure on auditory perception and the development of cognitive processing related to lexical tone discrimination.

Interestingly, the detection of pitch contour changes in natural speech (230–260 ms time window) was faster than that in English syllables (250–280 ms time window). This advantage in detection of pitch contour changes in native language syllables is consistent with the finding of Xi et al. that the MMN response to pitch changes in native syllables peaked earlier than that in nonspeech harmonics [23]. This effect might be attributed to the participants’ higher familiarity with Chinese syllables relative to English syllables. Specifically, the participants’ phonetic processing of the native syllables and familiarity with the NT stimuli might have led to faster detection of the difference in pitch contours.

While the present study provides insights into the development of the MMN response to Chinese lexical tonal contrasts and the transferability of phonological knowledge to nonnative pitch perception, several limitations should be acknowledged. First, the study’s participant groups were limited to specific age ranges (children aged 6–8 years, adolescents aged 12–14 years, and young adults aged 18–24 years). A larger and more diverse sample that includes a broader range of ages and linguistic backgrounds could provide more comprehensive insights into how lexical tone perception develops across different populations. Second, while the use of tonal chimeras allowed for controlled comparisons between native and nonnative syllables, future research could benefit from including nonspeech tonal contrasts or speech stimuli that more closely mimic natural speech variability. This could help determine if the observed results generalize to real-world speech perception scenarios. Third, although the use of t-PCA and component polarity realignment improved the quantification of the MMN response, potential limitations remain in the PCA approach, such as the choice of components and the subjective nature of component selection. Additional verification methods, such as cross-validation techniques, can be implemented to ensure the robustness of the findings. Furthermore, we propose adopting alternative dimensionality reduction and component analysis methods and implementing more automated approaches for component selection to reduce the inherent subjectivity in the process. These approaches might further strengthen the methodological rigor and enhance the reproducibility of our findings.

The observation that children can transfer their phonological knowledge of lexical tones to perceive pitch contours in nonnative syllables underscores the impact of linguistic experience on auditory processing. This suggests that early exposure to tonal languages may enhance general auditory processing abilities that extend beyond native speech. Understanding that phonological knowledge can be transferred to nonnative contexts may support the development of training programs that leverage a child’s existing language skills to facilitate learning of additional languages, especially those with different intonation or lexical tone structures. Future studies should explore the long-term impact of early tonal language exposure on auditory processing and cognitive development. It would also be valuable to examine how the development of MMN responses to tonal contrasts might influence other cognitive functions, such as memory and attention, and to assess whether similar patterns are observed in individuals with varying levels of exposure to tonal languages or various difficulties with linguistic prosody processing [44,71,72,73]. Additionally, investigating the effects of environmental factors, such as bilingualism or socioeconomic status, could provide deeper insights into the variability of MMN development and tonal perception across different populations.

## 5. Conclusions

In conclusion, our findings provide evidence that the MMN response to the Chinese lexical tonal contrast continues to develop and mature from childhood to adolescence, suggesting a gradual refinement of lexical tone perception capabilities over time. This neural development pattern indicates that the ability to detect and differentiate pitch changes, which is fundamental for recognizing tonal contrasts, is an ongoing process that evolves as children age and their auditory and cognitive systems mature. The observed differences in MMN amplitude between the child and adolescent groups further emphasize that the ability to process and respond to lexical tone contrasts becomes more robust during the transition from early to late childhood. Moreover, the phonological knowledge of lexical tones extends to the perception of pitch contours in both native and nonnative speech sounds from childhood.

## Figures and Tables

**Figure 1 brainsci-15-00093-f001:**
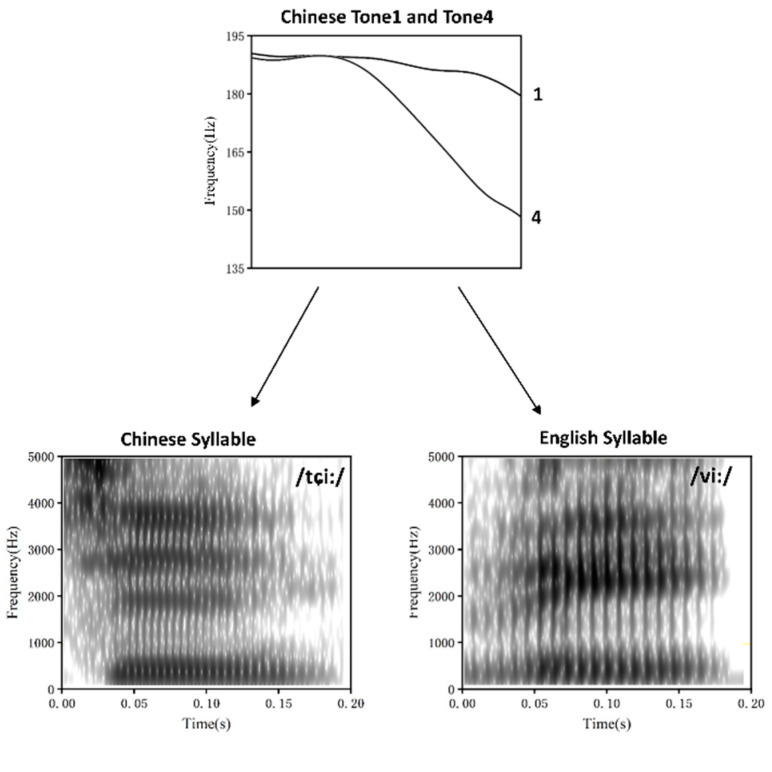
Schematic diagram illustrating how the two Chinese lexical tones (Tone1 and Tone4) are superimposed on the Chinese and English syllables (/tɕi/ and /vi/) to create Chinese and chimeric stimuli. The upper panel displays pitch contours of the two Chinese tones. The lower panel shows the broad-band spectrogram of the Chinese and English syllables.

**Figure 2 brainsci-15-00093-f002:**
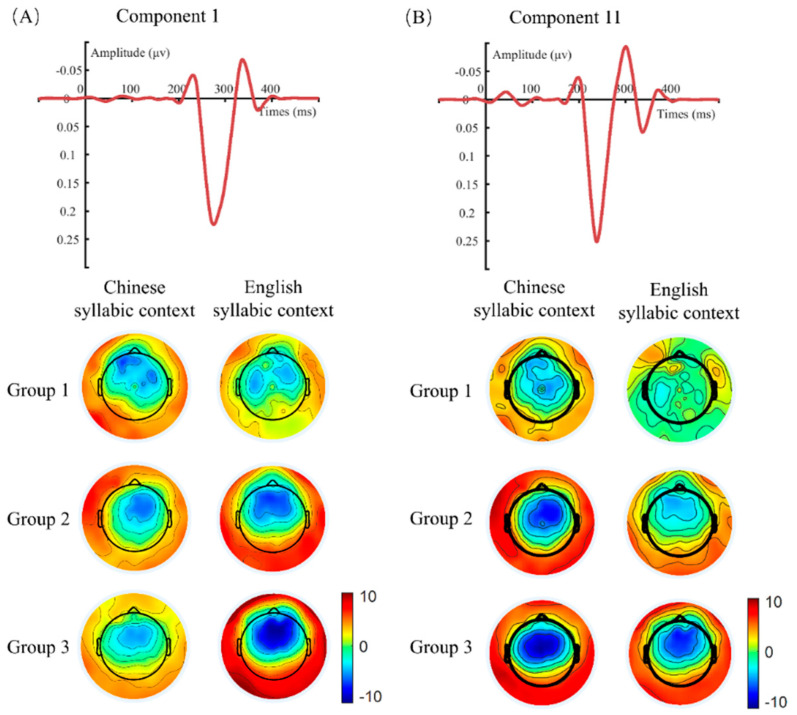
The waveforms and topographies of selected components. (**A**) The waveform and topography of component 1. Component 1 peaking at 274 ms explained 51.42% of the variance. (**B**) The waveform and topography of component 11. Component 11 peaking at 236 ms explained 1.14% of the variance. Each topography depicts the average scalp distribution of electrical potentials, with color gradients indicating neural activity magnitude and polarity. The map is characterized by a circular outline marking the reference plane, with the positions of the nose and ears providing a reference for the placement of electrodes.

**Figure 3 brainsci-15-00093-f003:**
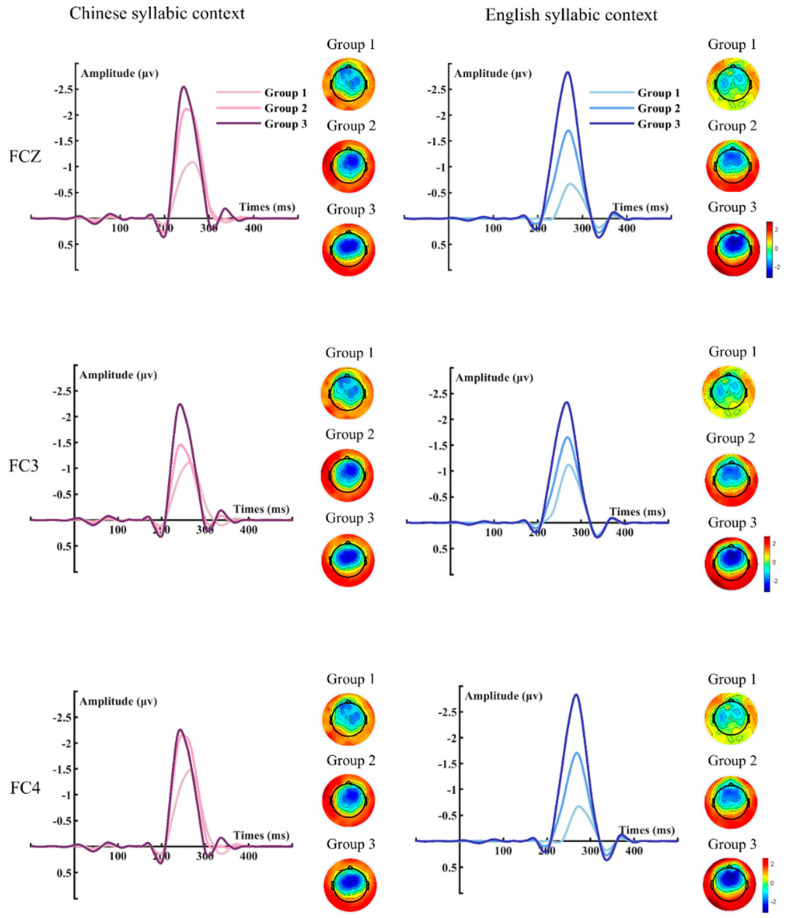
The average amplitudes and topographies of the three groups at Fcz, Fc3, and Fc4 electrodes for the Chinese syllabic and English syllabic contexts. The time window for the topography of Chinese syllabic context was set at 230–260 ms and the time windows for the topography of English syllabic context was set at 250–280 ms. Each topography depicts the average scalp distribution of electrical potentials, with color gradients indicating neural activity magnitude and polarity. The map is characterized by a circular outline marking the reference plane, with the positions of the nose and ears providing a reference for the placement of electrodes.

## Data Availability

The data that support the findings of this study are available from the corresponding author upon reasonable request due to restrictions (privacy, legal and ethical reasons).

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
