# Peer review of "The Neural Development of Chinese Lexical Tone Perception: A Mismatch Negativity Study Across Childhood, Adolescence, and Adulthood"

_brainsci, 2025, doi:10.3390/brainsci15010093_

Round 1
Reviewer 1 Report
Comments and Suggestions for Authors
The study focusses on demonstrating differences between children and adolescents/young adults in their perception of tonal contrasts in meaningfull and meaningless syllables, as well as in L1 (Chinese) as well as L2 (English).
Results clearly support the authors' thesis on the developmental acquisition of tone discriminations, with children (in the range of 6-8 years) showing significantly smaller amplitude than teens and young adults.
While amplitude results are very clear and well exposed, polarity ones appear to be less clear. Thus, my suggestion for an improved version of the manuscript is to better clarify the polarity section.
Some additional minor comments follow:
- Section 2.1: for trabsoarency, it would be better to add mean age of participants per group (+ s.d.).
- line 152: while keep should be "while keeping"
- line 229: a comma is missing (.. the MMN response, were selected.)
Reviewer 2 Report
Comments and Suggestions for Authors
The article provides valuable insights into the development of the MMN response to Chinese lexical tonal contrasts and the transferability of phonological knowledge to nonnative syllable contexts. However, there are areas where the presentation and depth of analysis could be improved to enhance the clarity and impact of the findings.
One area for improvement is the framing of the developmental trajectory of lexical tone perception. While the study effectively demonstrates that the MMN amplitude differs significantly between children and adolescents but not between adolescents and adults, the discussion could benefit from a deeper exploration of the underlying mechanisms driving this maturation. Linking the findings more explicitly to specific cognitive and neural processes, such as the development of executive functions or the refinement of auditory pathways, would provide a more nuanced understanding of why this developmental pattern emerges. Additionally, the discussion could address potential variability in developmental timelines across different populations and contexts, considering the role of individual differences and environmental factors such as socioeconomic status or bilingualism.
The discussion of the methodological implications of the study is thorough, but it could be expanded to better acknowledge its limitations and suggest directions for future research. For instance, the choice of age groups, while practical, restricts the ability to generalize findings to a broader developmental spectrum. Including participants from a wider range of ages, particularly younger children and older adults, could provide a more comprehensive picture of the developmental trajectory. Furthermore, the use of tonal chimeras, while innovative, may not fully capture the complexities of real-world speech perception. Future studies should incorporate stimuli that mimic natural speech variability to determine if the observed patterns extend to more ecologically valid contexts.
The discussion of the transferability of phonological knowledge is a strong point of the article, as it underscores the adaptability of linguistic experience in shaping auditory processing. However, the analysis could benefit from a more detailed examination of how early exposure to tonal languages influences broader auditory and cognitive processes. For example, the authors could explore how this transferability might extend to non-linguistic auditory tasks or how it interacts with cognitive functions such as attention and memory. These connections would situate the findings within a broader theoretical framework and emphasize their significance beyond the immediate context of tonal language perception.
The authors also acknowledge the limitations of their methodological approaches, particularly the use of t-PCA and component polarity realignment in quantifying the MMN response. While this is commendable, the discussion could go further by proposing specific methods to address these limitations in future studies. For example, employing cross-validation techniques to ensure the robustness of PCA-based findings would enhance the reliability of the results and provide a clearer pathway for replication and validation in subsequent research.
Lastly, while the article highlights the potential implications of early linguistic experience for language training programs, this section could be expanded to include more concrete applications. For instance, the discussion could explore how understanding the transferability of phonological knowledge might inform the design of educational interventions or language learning tools, particularly for children learning additional languages with tonal or intonation-based structures. Exploring the long-term cognitive and developmental impacts of early exposure to tonal languages could further enrich this practical perspective.
In conclusion, the article offers a significant contribution to our understanding of the development of MMN responses and the role of linguistic experience in shaping auditory perception. By expanding the discussion to include deeper theoretical insights, addressing methodological limitations with more specificity, and exploring broader applications of the findings, the study would have an even greater impact on the field of language development and cognitive neuroscience.
Reviewer 3 Report
Comments and Suggestions for Authors
I thoroughly enjoyed reviewing this manuscript, which examines the mismatch negativity (MMN) between Tone 1 and 4 across development, using a cross-sectional design. I only have a couple of suggestions for improvement.
First, it would be helpful to include a sample size justification. Were group sample sizes based on any sort of a priori power analysis, or based on prior work in this area?
Second, although the writing is generally quite clear, I would recommend removing language that comments on the importance or value of the work (e.g., "While the present study provides valuable insights..." Line 342). This should be determined by the readers of the work.
Round 2
Reviewer 2 Report
Comments and Suggestions for Authors
The revised version looks good! Thank you for your efforts!